# Implementation factors influencing the sustained provision of tele-audiology services: insights from a combined methodology of scoping review and qualitative semistructured interviews

Vidya Ramkumar,[1] Varsha Shankar  ,[1] Shuba Kumar[2]

[1]Department of Audiology, Sri Ramachandra Institute of Higher Education and Research (Deemed to be University), Chennai, Tamil Nadu, India
[2]Social Scientist, Samarth, Non-Government Organisation, Chennai, Tamil Nadu, India

**Correspondence to**
Dr Vidya Ramkumar;
vidya.ramkumar@sriramachandra.edu.in

## ABSTRACT

**Objectives** The objectives of the current study were to (a) identify long-term tele-audiology services reported to be implemented beyond the research phase and determine whether they are sustained, (b) map the implementation process to Standards for Reporting Implementation Studies guidelines and (c) map the factors that influenced its sustainability to the Implementation Outcomes Framework (IOF) to understand the gaps from an implementation research perspective.

**Study design, setting and participants** This cross-sectional study included a scoping review of articles describing long-term tele-audiology services from around the world to determine the factors influencing the implementation. Six electronic databases (PubMed, Cochrane Library, Web of Science, Scopus, Google Scholar and ProQuest) were searched for literature published between 2010 and 2023. This was followed by semistructured interviews (SSIs), which were guided by the IOF. Six project implementers were interviewed to obtain an in-depth understanding of factors that influenced sustainability of these tele-audiology services. Thematic analysis of the interview transcripts was carried out using a hybrid inductive-deductive approach.

**Results** Data were extracted from 32 tele-audiology studies included in the review, which were then mapped to 21 projects. The findings of the scoping review reveal that tele-audiology services were predominantly provided using synchronous telepractice methods. The 'professional-facilitator-patient' model was most commonly used. None of the studies reported the use of implementation research and/or outcome frameworks. Factors that influenced sustainability of tele-audiology services were identified from the combined results of the scoping review and the SSIs. These factors could be mapped to implementation outcomes of acceptability, adoption, feasibility, implementation cost and sustainability.

**Conclusion** Implementation research and/or outcome framework should be used to guide the implementation processes, its evaluation and measurement of outcomes systematically in tele-audiology service delivery. When such frameworks are used, gaps in information regarding the context influencing implementation, reporting of fidelity and adaptability measures can be addressed.

## STRENGTHS AND LIMITATIONS OF THIS STUDY

⇒ This study is guided by appropriate guidelines, for example, Preferred Reporting Items for Systematic Reviews and Meta-Analyses extension for Scoping Reviews, COnsolidated criteria for REporting Qualitative, Standards for Reporting Implementation Studies.
⇒ Appropriate frameworks were used to guide the scoping review (population-concept-context) and the semistructured interviews (SSIs) (Implementation Outcomes Framework).
⇒ Relevant data analysis methods (thematic analytical approach, a priori thematic saturation model, hybrid inductive-deductive approach) were used for the SSIs.
⇒ The limitations of the current study include the lack of a quality appraisal process.
⇒ The SSIs were conducted on only few implementers who were able to participate in the study.

## INTRODUCTION

Tele-audiology refers to the provision of audiology services using information communication technology. Tele-audiology has the potential to address the gaps between demand and needs, especially where access to specialised services is limited.[1 2] Tele-audiology has been explored over the last two decades for screening,[3 4] diagnosis[5–7] and rehabilitation.[8 9] The computerisation of audiology equipment has further expanded the applications of tele-audiology.[10] More recently, web portals and databases[11 12] have been developed to improve documentation and follow-up services in ear and hearing care. Tele-audiology can be conducted using synchronous mode, asynchronous mode, remote monitoring and mobile applications.[13] Several systematic and scoping reviews have examined telepractice applications for ear and hearing disorders

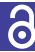

and the role of patient-site facilitators in tele-audiology service delivery.[14–18]

However, despite substantial research in tele-audiology, there is a gap between the positive research outcomes and its sustainable implementation in the real-world setting. Integration of evidence-based interventions into routine clinical and community settings requires adoption of suitable implementation strategies, which forms the foundation of implementation research.[19–23] Further, in order to achieve sustainability, outcomes suitable to the stage of implementation must be periodically monitored to make suitable adaptations.[22] Implementation research and outcome frameworks help achieve translation of research evidence to sustained implementation.[24] For example, the Implementation Outcomes Framework (IOF) provides a model of eight conceptually distinct implementation outcomes.[22] However, in general, implementation of allied health services has not been guided by implementation research.[25]

Particularly with regard to tele-audiology services, there is a lack of data on sustained implementation efforts and the factors that influenced their sustainability. As a result, the current study's objectives were to (a) identify long-term tele-audiology services reported to be implemented beyond the research phase and determine whether they are sustained, (b) map the implementation process to Standards for Reporting Implementation Studies (StARI) guidelines and (c) map the factors that influenced its sustainability to IOF[22] to understand the gaps that must be addressed from an implementation research perspective.

## METHODS
### Patient and public involvement
Patients or the public were not involved in the design, or conduct, or reporting, or dissemination plans in the current study.

This study included a scoping review followed by a qualitative component involving semistructured interviews (SSIs). The scoping review was carried out to identify long-term tele-audiology services reported to be implemented beyond the research phase. The scoping review was also used to map the implementation process to StARI guidelines. SSIs were used to determine whether or not these services are sustained. The findings from the scoping review and the SSIs were used to map the factors that influenced its sustainability to IOF.[22]

### Scoping review
The Preferred Reporting Items for Systematic Reviews and Meta-Analyses extension for Scoping Reviews guidelines (PRISMA-ScR)[26] was used to develop the scoping review protocol (online supplemental file 1). The population-context-concept framework was used to formulate the eligibility criteria which included search terms, strategies, quality appraisal criteria and extraction methods. Search strategies were finalised after an initial pilot was carried out (online supplemental file 2). An initial decision

profile (online supplemental file 3) was designed and created for systematic documentation of the selection of articles.

Six electronic databases (PubMed, Cochrane Library, Web of Science, Scopus, Google Scholar and ProQuest) were searched. The Rayyan software[27] was used for duplicate removal and article selection process. Two reviewers were involved during this entire process, and any disagreements between the two were discussed until a consensus was established.

Studies describing tele-audiology service provision (including screening, diagnostic, or rehabilitative services) to individuals of all age groups, based on prior research evidence were included. Studies describing 'long-term' or 'routine' or 'sustained' tele-audiology service delivery, for 2 or more years, were only included. Quasi-experimental trials, and community or field trials, were also included. All telepractice modalities, including video conferencing, web-based, mobile applications and remote computing, were considered. Only literature published in English between 2010 and 2023 was considered.

Studies that compared in-person and telepractice measures, or reported validity of tests or tools but did not report or mention long-term implementation were excluded. When full texts were not available after reasonable efforts, articles were excluded.

### Analysis
The search results were summarised using the PRISMA flow diagram. We extracted information on geographical distribution, the focus area of service delivery, method and model of tele-audiology service delivery. The StaRI guidelines[28] was used to map the implementation strategies and components reported in the studies (online supplemental file 4). When a StARI criteria was addressed, it was denoted with a check mark, when it was partially addressed, it was denoted with a '?', and when it was not addressed, it was denoted with a 'X.' When a criterion was not relevant, it was marked with 'N/A.' The contents of the articles were examined for reports of factors that influenced the sustainability of tele-audiology services. Any recommendations by the implementers were also considered as influencing factors (online supplemental file 5). These were then mapped to the constructs of the IOF.[22]

### Semistructured interviews
The long-term and sustained tele-audiology service implementers identified in the scoping review were included for the SSIs. This was done to assess if these services are still sustained and also exemplify the data obtained from the scoping review on the factors that influenced the sustainability of these services. The COnsolidated criteria for REporting Qualitative research guidelines[29] were used to report the qualitative components of the study (online supplemental file 6).

### Research team
#### Personal characteristics and relationship with participants
Both VR and VS conducted three interviews. VR has 10 years of experience performing qualitative research; VS had no prior experience. There was no specific relationship established by the interviewers with the participants prior to the interview. Information about the interviewer's interest in the research topic, as well as purpose and description of the research study, was provided in the e-consent form.

### Study design
#### Theoretical framework and development of interview guides
The thematic analytical approach is useful to identify, analyse and report patterns in the data obtained[30 31]; thus, in order to better understand the factors that influenced the sustainability of the tele-audiology services across different implementers, we used this approach as our methodological orientation for this study. The interview guides were developed based on the constructs of the IOF[22] to which the influencing factors could be mapped to. It was then verified by two experts with experience in qualitative research and telepractice. The guides were verified for content, wording and relevancy to objectives. A copy of the interview guide has been provided (online supplemental file 7).

#### Participant selection
Sampling frame included for the SSIs were the primary and corresponding authors of all the papers identified in the scoping review. The corresponding authors were first contacted; if they were unable to participate, we requested a referral to any of the other authors of the team. If we were unable to establish contact with the corresponding author, then an effort was made to contact other authors.

Emails were sent to all the authors along with the link to an e-consent form. In a couple of instances, the authors led the investigators to another team member. Reminders were provided for scheduling the interviews and the consent form remained available for 3 months. Out of the 21 authors identified in the scoping review, 6 authors in their early, mid and advanced career levels, consented to participate in the study. The participant characteristics are described in table 1.

### Setting
Since interviewees belonged to different countries around the world, all interviews were conducted using an encrypted online video-conferencing platform. Other than the investigators, no one else was present during the interview.

### Data collection
The interview guides were initially piloted on a project implementer and suitable adaptations were made prior to data collection. Primary guides and additional probes were used to obtain rich data. The interview guides were further informed and expanded on based on each interview, thereby enabling deeper insights.

Informed e-consent was obtained from all the participants prior to the interviews. All interviews were audio-video recorded and field notes were also taken by the interviewers. All interviews were completed within 30–45 min. The 'a priori thematic saturation'[32] model was used in which the adequacy of data collected was determined when informational redundancy across interviews was apparent.

### Analysis
#### Data analysis
We conducted thematic analysis of verbatim transcription of all the audio and video recorded interviews using NVivo V.12 software. Thematic analysis was carried out using a hybrid inductive-deductive approach by reading through the transcripts and looking for patterns to identify themes.[31 33] After familiarisation with the transcripts, both the authors coded two transcripts independently and updated the existing codebook by adding codes generated deductively from the interviews. Any coding differences were discussed and resolved. On fine-tuning, the remaining transcripts were coded based on the updated codebook. Unique and relevant information that were

| Code | Description of participant | Gender | Experience levels | Country |
|------|---------------------------|--------|-------------------|---------|
| 01 | Project implementer of a long-term telepractice service delivery programme | Male | Advanced | USA |
| 02 | Project implementer of an ongoing telepractice service delivery programme | Female | Mid-career level | USA |
| 03 | Project implementer of an ongoing telepractice service delivery programme | Male | Mid-career level | Poland |
| 04 | Project implementer of a community-based telepractice service delivery programme | Female | Early-career level | India |
| 05 | Project implementer of a community-based telepractice service delivery programme | Male | Early-career level | India |
| 06 | Project implementer of a hybrid telepractice service delivery programme | Female | Early-career level | South Africa |

**Table 1** Description of SSI participants

SSIs, semistructured interviews.

identified during this process were given new codes. We organised the codes based on similarity and frequency, which aided in categorising codes. We analysed these categories for important patterns in the data that were relevant to our study objectives and compiled them under themes. Following this analysis, we evaluated the themes for relevancy and credibility. We merged certain themes together for a more succinct representation of the findings whenever necessary.

### Reporting

We presented an explanation of our study findings, which were substantiated with quotes placed under the respective themes with specific qualifiers for each participant.

All quotes and findings were sent back to the interviewees for comments and corrections.

## RESULTS

The results describing, project mapping, geographical distribution of projects, focus area of service delivery, method of telepractice, model of telepractice service delivery and mapping of implementation components to the StaRI guidelines are based on the findings from the scoping review. The section addressing the factors that influenced the sustainability of tele-audiology services include findings of both the scoping review and the SSIs.

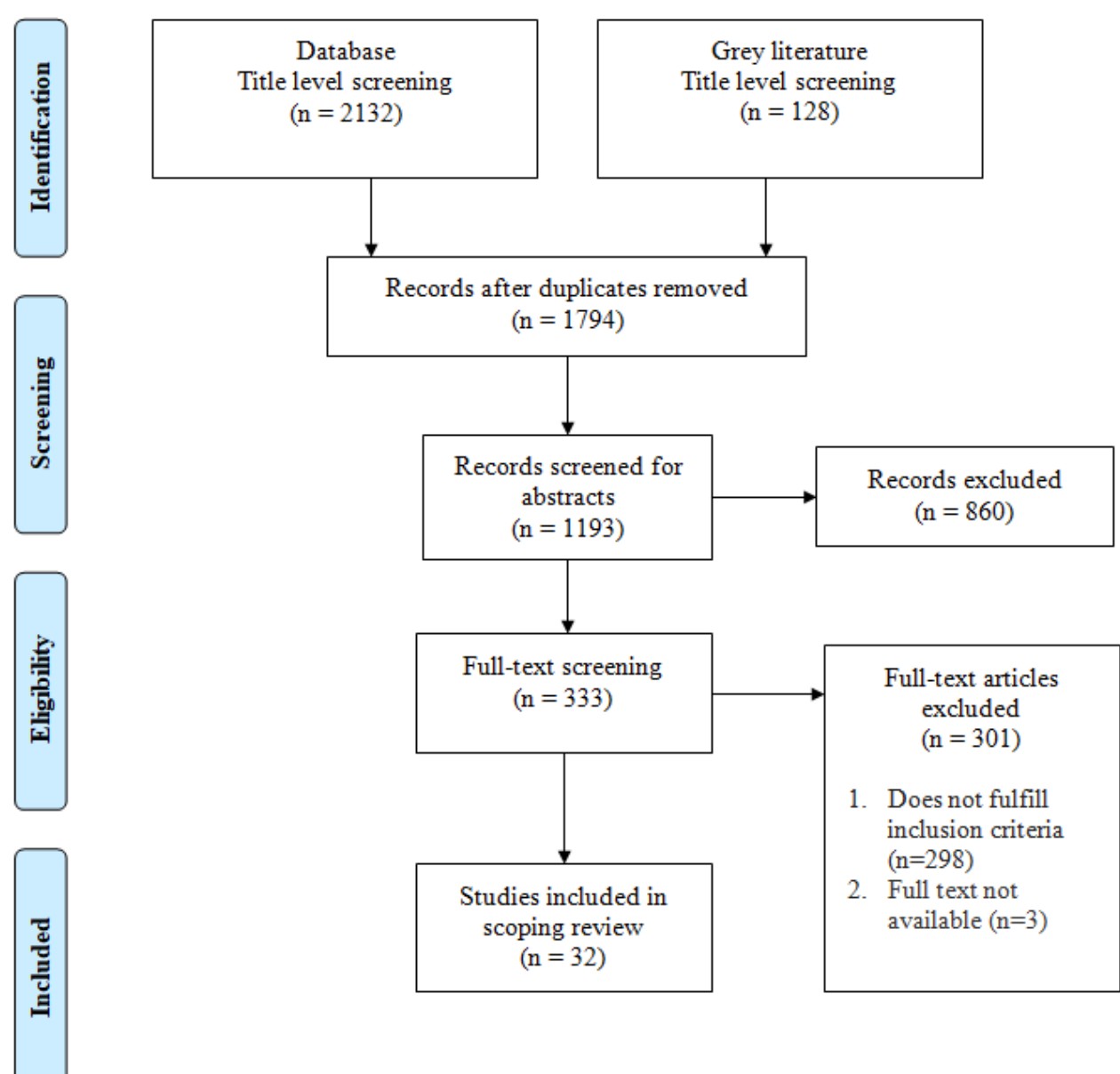

**Figure 1** PRISMA flow diagram representing the study's search process. PRISMA, Preferred Reporting Items for Systematic Reviews and Meta-Analyses.

**Table 2**  Summary of project mapping in tele-audiology

| Focus area | Country | Project | Project code |
|---|---|---|---|
| Hearing screening | USA | Emmett et al[40]<br>Emmett et al[41]<br>Robler et al[42] | P1 |
| Diagnostic audiological evaluation | Canada | Dharmar et al[53] | P2 |
| | | Stuart[43] | P3 |
| | Canada | Campbell and Hyde[44] | P4 |
| | | Hatton et al (2019a)[36] | P5 |
| | India | Ramkumar et al[37] (2018a)<br>Ramkumar et al[34] | P6 |
| Diagnostic audiological with otorhinolaryngological evaluation | Canada | Hofstetter et al[49]<br>Kokesh et al[50] | P7 |
| | Australia | Smith et al[51]<br>Smith et al[35]<br>Smith et al[74] | P8 |
| | India | Gupta et al[54] | P9 |
| | | Ravi et al[38] | P10 |
| Hearing aid fitting and programming | USA | Dennis et al[45] | P11 |
| | | Novak et al[46] | P12 |
| Cochlear implant fitting and mapping | USA | Luryi et al[55] | P13 |
| | Poland | Skarżyński et al[39]<br>Skarżyński et al[56] | P14 |
| Aural re/habilitation | USA | Houston[75]<br>Blaiser et al[58]<br>Houston and Stredler-Brown[57] | P15 |
| | | Broekelmann[59] | P16 |
| | | Lalios[60] | P17 |
| | | Galvan et al[76] | P18 |
| | Australia | McCarthy et al[62]<br>McCarthy[47]<br>McCarthy[52] | P19 |
| Comprehensive audiological services | USA | Gladden et al[48] | P20 |
| | South Africa | Ratanjee-Vanmali et al[77] | P21 |

The scoping review identified 32 articles that satisfied the inclusion criteria. The PRISMA flow diagram depicts the studies search results (figure 1). A summary of the literature reviewed based on the scoping review's inclusion criteria was tabulated for analysis. The study code, study title, authors, countries and focus area are all listed in table 1a in online supplemental file 8. The settings, participants, employees and types of services are described in table 1b in online supplemental file 8.

## Project mapping

Studies describing similar background content in the introduction, common implementation sites, broad objectives and a contiguous group of investigators were mapped as a single project. This exercise was undertaken to make a realistic assessment of the implementation focus. Studies were considered independent projects when they could not be mapped under any other larger project. Twenty-one projects were identified based on this mapping exercise (table 2) and each project was coded as P (project) and serial number, for ease of reference. The subsequent findings will be reported using these project codes.

## Geographical distribution of projects

Tele-audiology was implemented in 12 projects in the USA; 2 projects in Australia; 3 projects in India; 2 projects in Canada; 1 project each from Poland and South Africa. Projects implemented in almost all the countries (P1–10; P14; P16; P18–20) were delivered to overcome access barriers for individuals in rural areas and/or remote locations. However, in the USA, there were six projects (P11–13; P15; P17; P21) that provided tele-audiology services to urban and semiurban areas due to scheduling related issues (work, having other children to take care of, household responsibilities).

## Focus area of tele-audiology service delivery

Tele-audiology was implemented in projects with a focus on ear and hearing screening, diagnostic audiological evaluations, a combination of diagnostic audiological with otorhinolaryngological evaluation, hearing aid



fitting and programming, cochlear implant fitting and mapping as well as intervention services.

Mobile phone-based ear and hearing screening was carried out among school-aged children (P1). Teleprac-tice was used to conduct diagnostic confirmation of hearing loss among infants in follow-up to hearing screening (P2–6). Tele-auditory brainstem response (ABR) and teledistortion product otoacoustic emissions (DPOAEs) were performed in all these projects. Other diagnostic tests such as immittance including high-frequency tympanometry and middle ear muscle reflexes, and auditory steady-state response (ASSR) were included only in a few projects (P2, P3).

Diagnostic audiological evaluation combined with otorhinolaryngological evaluation using telepractice was reported in four projects (P7–10). In these projects, basic ear (video-otoscopy) and hearing testing (audiometry and tympanometry) was provided as a routine service to individuals of all age groups.

Remote hearing aid fitting, verification and program-ming (P11, P12) as well as remote cochlear implant switch-on and mapping (P14) were conducted for all age groups. One project focused on cochlear implant mapping for geriatric patients alone (P13). In the projects where remote cochlear implant switch-on and mapping were conducted (P13, P14), tests such as acoustic reflex threshold/neural response telemetry/neural response imaging and e-stapedial reflex threshold was carried out through telepractice. Telepractice was used for aural re/habilitation of both children and adults (P15–19) to provide auditory-verbal therapy or some form of habilita-tion training to individuals with hearing loss.

Comprehensive audiological services to individ-uals of different age groups were provided with tests including audiometry, hearing aid fitting and program-ming, cochlear implant fitting and mapping in one project (P20). On the other hand, another project (P21) conducted only hearing screening and hearing aid repro-gramming using telepractice, for adults. The diagnostic confirmation and hearing aid fitting was done in person.

## Method of telepractice

Use of synchronous/real-time, asynchronous/store and forward, or a combination/ hybrid method for providing tele-audiology services was identified from the scoping review.

In tele-audiology, the asynchronous method of service delivery was predominantly used for audiometric screening, video-otoscopy, and tympanometry (P1; P7–10). Client details, video-otoscopic images and audio-metric or tympanometry data were stored and forwarded to an audiologist or otolaryngologist for review. For tele-diagnostic evaluation of hearing among infants, synchro-nous methods (for both ABR and DPOAEs) or hybrid methods of synchronous (for ABR) and asynchronous (for DPOAEs) were used (P3–6). In one project (P2) all diagnostic tests (video-otoscopy, middle ear analysis, DPOAEs, ABR and ASSR) were completed synchronously.

Remote hearing aid fitting and programming (P11, P12, P20, P21) and remote cochlear implant fitting and mapping (P13, P14), were conducted synchronously. In one project (P12), even real-ear probe measures were conducted synchronously. Aural habilitation services such as auditory verbal therapy (AVT) were provided synchro-nously (P15–19).

## Model of telepractice service delivery

When an e-helper/telepractice assistant/support staff supported patient care, it was categorised as 'professional-facilitator-patient' model; when the professional delivered the service directly to the patient without intermediary personnel, it was categorised as 'professional-patient' model; and 'professional-professional-patient' model involved a second audiologist or any other professional at the patient site.

In tele-audiology, irrespective of test procedures, the 'professional-facilitator-patient' model was used whenever services were provided to remote rural areas. In this case, a trained e-Helper/telepractice assistant/community-based health worker served as a facilitator for telepractice (P1–6; P8–12; and P20) and their roles were defined in the project. This was the most commonly used model of telepractice service delivery. The 'professional-patient' model was predominantly used for services provided through video-conferencing such as AVT and habilita-tion of individuals with hearing loss (P15–19; P21). The 'professional-professional-patient' model was used for fitting cochlear implants (P13, P14) when both audio-logical and otorhinolaryngology services were provided (P7). The professional at the patient's end was an audi-ologist with no specialised training in cochlear implant aural rehabilitation, while the professional who provided services via telepractice was specialised in this area (P13). In one project (P14), a speech therapist was the professional available at the patient end to facilitate the tele-fitting of cochlear implants. Otolaryngology consul-tations were supported at the patient end by a variety of professionals including audiologists, physicians, public health nurses, dentists and physician assistants, as a part of a routine telemedicine clinic (P7).

## Mapping of the implementation process to StaRI guidelines

We found that none of the long-term projects used any implementation framework to implement tele-audiology service delivery. Thus, we mapped the implementation process described in the studies to the StaRI guidelines. Only two study titles[34 35] and six study abstracts[34–39] iden-tified their study as 'implementation' research. None of the studies mentioned the use of an underlying theory/framework/model or relevant reporting guidelines. Eleven studies provided a detailed overview of the context influencing implementation.[34 36 40–48]

The outcomes of the implementation strategy were spec-ified in 14 studies.[34 36 38 40–42 45–52] While 18 studies included process evaluation objectives,[36 39–42 46–48 50 52–60] only 12 studies reported the process outcomes.[36 39 46–48 50 52 54–56 58 60]

The economic evaluation of the implementation strategy was described only in seven studies.[34 36 38 45 49 50 61] Although description of fidelity to the planned implementation strategy and contextual adaptations was reported in four studies,[36 42 48 54] specific measures to assess fidelity and adaptation were not described. Fourteen studies each addressed scalability[36 37 39 42 44–48 52 56 58 59 62] and sustainability[36 37 39 44–48 52 55 58–60 62]

### Mapping of the influencing factors to the IOF

The results of the scoping review (denoted using the project codes) and thematic analysis of the SSIs (substantiated using quotes) were combined to map the factors that influenced sustainability of tele-audiology services. The quotes are summarised (online supplemental files 9) under the respective overarching themes with specific qualifiers for each project implementer.

We found that none of the projects studied implementation outcomes systematically. However, studies highlighted the factors influencing the sustainability of tele-audiology service delivery based on their experiences. These reports could be mapped to the implementation outcomes of acceptability, adoption, feasibility, implementation cost and sustainability.[22]

### Acceptability

The beneficiary/patient perception related factors reported by the project implementers could be mapped to the early to late-stage implementation outcome of acceptability. Parents were hesitant to adopt tele-audiology in the beginning as they were apprehensive about the child's ability to build rapport with the therapist. In such a case, they were given a choice of a combined approach of in-person and telepractice sessions until they felt comfortable and confident (P5, P19) or continue in-person sessions till they expressed their willingness to engage in telepractice sessions (P15, P18). The families were provided with the opportunity to meet the therapist and participate in a practice video conference session before the commencement of telepractice sessions. Such a mindful approach was found to help address patient apprehensions (refer to quote 1). Parents also became ambassadors of the programme when they saw the value in it (refer to quote 2).

Patients' or caregivers' perceptions and understanding of the advantages of tele-audiology service delivery was a key facilitator (P5, P7, P8, P14, P15, P19, P20) (refer to quote 3). Continuous and ongoing engagement with communities in a culturally safe manner and building awareness regarding the service enhanced acceptance (P1, P8) (refer to quotes 4 and 5). Reduced travel cost associated with tele-audiology services was perceived as facilitatory for implementing such services (P12, P14). Real-time presence of the professional, technical team and support staff and reassurance by them, facilitated the acceptance of tele-audiology by patients/caregivers (P15, P18, P19–20). Patient's acceptance of technology use in ear and hearing care improved when visual images

of tests conducted (image of tympanic membrane) were provided (refer to quote 6). Acceptance was also conditional to the absence of any direct cost to the patient (refer to quote 7) or in case of subsidised service (refer to quote 8). Patients felt secure when data security, privacy and confidentiality were protected (P5, P16). Participation in tele-audiology sessions by other family members, who otherwise would not be able to do so, was further reassuring in some instances (P15, P19). Setting appropriate expectations for the patient and ensuring clear communication prior to telepractice service delivery can enable acceptance (refer to quote 09).

### Adoption

Provider perspectives regarding uptake and utilisation of tele-audiology service delivery could be mapped to the early-stage implementation outcome of adoption. Project implementers felt that the assessment of a patient's suitability for tele-audiology was crucial and it was important to clearly communicate to the patient when telepractice might not be the best option (refer to quote 10). Adoption of new service delivery requires change and this change needs to be supported and driven by the leadership with a vision for sustainability (refer to quote 11). Organisations see value in telepractice because of their threefold advantage: teaching, research and service delivery (refer to quote 12). Hesitancy among service providers to adopt tele-audiology was considered as a challenge to implementation (refer to quote 13).

### Feasibility

Factors related to the utility and practicality of implementing tele-audiology service delivery could be mapped to the early-stage implementation outcome of feasibility. Assessing needs, availability of internal resources, financial planning and a service delivery model that is flexible and adaptable to change was considered useful for seamless implementation of tele-audiology services (P9, P20) (refer to quote 14). Formulating a memorandum of understanding and a telehealth service agreement before implementation was reported to be facilitatory (P5, P11). Integrated electronic health records were also recommended to ease tele-audiology service implementation (P7, P20).

Limited or non-availability of professionals and technicians trained in tele-audiology service delivery was reported to be a major barrier to implementation (P15). Retention of personnel employed in tele-audiology service delivery was a challenge as it was perceived to be labour-intensive (refer to quote 15). The uniqueness of telepractice draws personnel to work in such services; however, continuity is supported by monetary benefit, opportunities for career development and positive outcomes of services (refer to quote 16). In providing remote care, establishing a local 'point of contact' who is dedicated and committed to assisting telepractice service delivery was beneficial (refer to quote 17). Having a liaison from the community helped minimise and overcome cultural barriers (refer to

quote 18). A dedicated team including professionals, and administrative and technical support staff was considered pertinent (P6, P19) (refer to quote 19). Reliable patient outcomes were associated with team-building, coordination and streamlined communication between all collaborators involved in telepractice service delivery (P5, P11, and P13) (refer to quote 20). Further, defining the roles and responsibilities of e-helper/ telepractice assistants with systematic culture-sensitive training was considered useful (P7, P20) (refer to quote 21).

Devices with a simple user interface, minimal skill requirement, display in local language and adequate battery back-up were essential to support remote community-based tele-audiology services (refer to quote 22). Poor internet bandwidth compromised service delivery in rural areas (P10, P15, P17) (refer to quote 23) and the use of undersea cables and broadband connections were regarded as more reliable (P5, P19) (refer to quote 24). High-quality dedicated video conferencing equipment was a quality enhancer in tele-audiology service delivery (P14, P19, P20). In rural areas where issues in connectivity were expected, the store and forward method of tele-audiology was found beneficial (refer to quote 25).

Capitalising on available resources (patient's own devices, pre-existing infrastructure) facilitated seamless implementation (refer to quotes 26 and 27). Contrary to popular belief, digital proficiency was not reported to be a significant barrier to implementation, as it was felt that a supporting mechanism can be created (refer to quote 28).

Lack of acts, laws, malpractice liability and credentialing issues were also some of the reported challenges to tele-audiology service implementation (P11). In certain countries where state licensing was essential, unique licensing policies to deliver telepractice services across states was required (refer to quote 29). Some of these legislative policies gained popularity only after the COVID-19 pandemic and project implementers felt that these changes needed to be made permanent (refer to quote 30).

## Implementation cost

Factors related to cost of setting-up or cost associated with continued tele-audiology services could be mapped to implementation cost which is an early to late-stage implementation outcome. Higher initial capital investment to set up tele-audiology service delivery and the need for specialised equipment to support such practice was reported as a barrier to implementation (P6, P12, P15 and P20) and was only possible with adequate funding (refer to quote 31). When there was a pause or temporary issue with the release of funds, it adversely affected service delivery (refer to quote 32). Even though tele-audiology has gained popularity and preference in the pandemic, it was not adequately supported financially by organisations in the past for routine service delivery (refer to quote 33). Multiple sources of funding including scientific grants,

organisational funding, and charitable donations were required to ensure continuity of service (P19, P20) (refer to quote 34). On some occasions, lack of insurance reimbursement for tele-audiology services was a barrier (P11) (refer to quote 35).

When long-term cost outcomes or cost benefits of tele-audiology were evaluated, they found it to be a beneficial alternative from a societal perspective (P10, P15). Considerable efforts were taken to establish and self-sustain billable tele-audiology service delivery and these were very few (P21) (refer to quote 36).

## Sustainability

Factors reporting the maintenance of tele-audiology service delivery were mapped to the late-stage implementation outcome of sustainability and these were minimal. Out of the identified projects in tele-audiology, 14 projects (P1–5, P7, P9–11, P13, P14, P18–20) reported service delivery to be ongoing or routine. The rest, even though long term, were still in the research phase. Tele-audiology services were implemented in a more sustained manner in hearing screening programmes for infants. For example, follow-up diagnostic evaluations were conducted using telepractice in the USA and Canada (P2–5), ear and hearing screening among school children was conducted in the USA (P1) and diagnostic audiological with otorhinolaryngological evaluation for adults was conducted in the USA and in India (P7, P9, P10). The Veterans hospitals in the USA have implemented remote hearing-aid fitting and programming as well as cochlear implant fitting and mapping as a routine service (P11, P13 and P20). Another effort is the National Network of Teleaudiology in Poland for cochlear implant switch-on and mapping (P14). Aural rehabilitation has also been implemented in a sustained manner in both the USA and Australia (P18, P19). Systematic preimplementation feasibility, test runs and validation of functional requirements were useful components of sustainable implementation (P9, P5 and P18). Using existing resources to ensure implementation even after the research phase was considered facilitatory (refer to quote 37). Continued support from the organisation was considered crucial for sustainable telepractice service delivery (refer to quote 38). The lack of interlinking support structures across administrative health authorities was reported to be detrimental to sustainability (P5).

## DISCUSSION

Through this study, we identified tele-audiology programmes that provided long-term or sustained services for at least 2 years. We identified the factors that influenced their sustainability which we then mapped to the IOF[22] to understand the gaps that must be addressed for improved translation of research evidence to sustained implementation.

In general, we found that tele-audiology was predominantly implemented in remote/rural locations to

overcome practical challenges associated with receiving in-person services. The most common application of tele-audiology was for diagnostic testing with ABR or OAE in infant hearing screening programmes. This was done to reduce the lost to follow-up especially when hearing screening was conducted in primary healthcare clinics or rural communities.[36 37 43 53] Aural re/habilitation for hearing loss was the next most common application, where only simple video-conferencing software was required. Remote hearing aid fitting and programming was minimally explored, possibly due to the limited availability of supportive features in products of a wider price range.[14] There is a definite preference for using synchronous methods for conducting diagnostic tests or fitting hearing devices when audiologists' real-time judgement is more crucial.[63] Whenever screening was done or images could be captured (like video-otoscopy or tympanometry), asynchronous methods were used. In such tests, real-time assessments by the professional were not considered to be critical for test completion. It was possible to provide telepractice services, such as aural re/habilitation, at homes using simple real-time videoconferencing which did not mandate specialised skills.[64] In these cases, patient preparation was necessary and the parent/caregiver acted as a facilitator.

None of the studies included in the current review reported the use of an underlying implementation theory/framework/model and implementation outcomes were not studied systematically. More than half of the studies assessed process evaluation objectives, which were mostly related to patient satisfaction, but only a smaller subset reported its outcomes. In addition, only a few studies reported economic evaluations. We identified two projects (P1, P7) where tele-audiology services were provided as a part of comprehensive multispecialty services. Impact evaluations of such implementations are likely to inform the value of capitalising on common infrastructure and shared human resources. The core component of the intervention (tele-audiology service) to which fidelity is expected and those aspects that may be adapted must be systematically reported to aid implementation. But, none of the studies reported the use of fidelity or adaption measures.

While the use of implementation research and/or outcome frameworks for translating evidence-based practice to clinical service delivery is gaining momentum, it is still underused in tele-audiology service implementations. Only one recent study reported the use of implementation science framework to investigate the feasibility outcomes of a 17-week tele-audiology hearing aid service.[65] Similar efforts for long-term implementation will offer deeper insights on sustainability aspects.

The factors reported to influence the sustainability of these services were mapped to the constructs of IOF. These factors were predominantly related to early-stage implementation outcomes of feasibility and early to late-stage implementation outcomes of acceptability. Limited factors could be mapped to the implementation outcomes of adoption (early stage), implementation cost (early to late stage) and sustainability (late stage).

With reference to feasibility, internet access is a basic prerequisite for tele-audiology. The internet's speed and bandwidth affect various aspects of tele-audiology implementation, including test selection, technique and model of service delivery.[66] The use of broadband connectivity and undersea cables was reported to be useful by implementers, yet there seems to be a continued challenge in maintaining robust connections for the purpose of tele-audiological tests in remote/rural locations.[67] In addition, legislative policies and guidelines influenced feasibility of implementation considerably. Several countries introduced technical guidelines and changes in legislative policies for telepractice only during the COVID-19 pandemic.[68] However, supportive mechanisms such as integrated electronic health records are still not widely available at a national level.[69]

Acceptability and perceived demand for tele-audiology services by both the patients/beneficiaries and the providers seems essential, as they are primary stakeholders involved. The findings of the current study indicate that better administrative convergence is required between health authorities for implementation of tele-audiology services. The provider's willingness and comfort in adapting their services using technology is crucial for sustainability of tele-audiology service delivery.[70]

For adoption of tele-audiology within organisational structures, supportive organisational policies that recognise it as an important component of service delivery is required.[71] This will in-turn impact various organisational aspects including the availability of human resources, collaborations and effective planning. Tele-audiology is unlikely to be sustained without a clear plan that defines roles and responsibilities.[72]

With reference to implementation cost, our findings suggest that even though tele-audiology services were initiated through research grants, several of them were able to integrate these services into public-funded programmes. This suggests that sustained implementation occurred when the need and usefulness of tele-audiology were accepted by public sector policy-makers. If research projects include economic evaluation, then this can in turn inform the allocation of funds by organisations to support sustained implementation.[73]

Factors related to appropriateness, fidelity and penetration, which are early, mid and late-stage implementation outcomes, respectively, were not reported. Information regarding the context influencing implementation, the outcomes of the implementation strategy, process evaluation objectives, process outcomes, economic evaluation, specific measures to assess adaptation and scalability were not described. When appropriate implementation research and outcome frameworks are used, the stages or processes involved in achieving sustainability are less likely to be missed.



The strength of the current study lies in the representation of tele-audiology implementation efforts in a wide range of countries that include low-middle-income (India and South Africa) and high-income (USA, Canada, Australia and Poland) nations. We found that the factors influencing the sustainability of tele-audiology service delivery reported in both high-income countries (HICs) and low-income and middle-income countries (LMICs) were similar, except with respect to insurance coverage for tele-practice, and the availability of electronic or digital health records which emerged in the context of HICs and not LMICs. Thus, we believe that the findings of this study will be useful to tele-audiology project implementers from different regions.

There are certain limitations which could influence the scope of the current review. The study lacked a quality appraisal process and there is a possibility that articles published in non-native English-speaking languages could have been missed. Only six tele-audiology project implementers consented to participate in the SSIs, resulting in a small sample size.

**Acknowledgements** We would like to thank all the researchers who consented to participate in the interviews sharing their valuable time for furthering the understanding in this area of research.

**Contributors** VR is the gurantor for the overall content of this manuscript, VR conceptualised the research topic, both VR and VS were involved in data curation, formal analysis, investigation, finalising methodology, original draft preparation and reviewing and editing. VR was also integral in supervision, visualisation and acquisition of funding, resources and software. Both authors read and approved the final manuscript. SK was involved in methodology for coding, thematic analysis and manuscript review and editing.

**Funding** The authors disclosed receipt of the following financial support for the research, authorship and/or publication of this article: This work was supported by the DBT/Wellcome Trust India Alliance Fellowship/Grant (IA/CPHI/19/1/504614) awarded to Vidya Ramkumar.

**Competing interests** None declared.

**Patient and public involvement** Patients and/or the public were not involved in the design, or conduct, or reporting, or dissemination plans of this research.

**Patient consent for publication** Not applicable.

**Ethics approval** This study involves human participants and ethical clearance was obtained from the Sri Ramachandra Institute of Higher Education and Research (DU) Institutional Ethics Committee (reference number: CSP/20/NOV/87/191) for this study. Participants gave informed consent to participate in the study before taking part.

**Provenance and peer review** Not commissioned; externally peer reviewed.

**Data availability statement** Data are available on reasonable request. The data that support the findings of this study are available on request from the corresponding author, VR. The data are not publicly available as the data underlying this study consists of transcripts of the qualitative interviews that identify the participants and cannot be sufficiently deidentified. To ensure confidentiality and data protection, we have only included relevant excerpts from the interviews with 'no objections' from the participants. We will only provide the transcripts to researchers/those that provide a proposal of how the data will be used by them. The corresponding author will ensure to remove identifiers from the particular data set and obtain 'no objections' from the participants before sharing the same.

**ORCID iD**
Varsha Shankar http://orcid.org/0000-0002-3633-6230

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
