## [Reviewer comments · BMJ Open]

ARTICLE DETAILS

TITLE (PROVISIONAL)	Implementation Factors Influencing the Sustained Provision of Tele-Audiology Services: Insights from a Combined Methodology of Scoping Review and Qualitative Semi-structured Interviews
AUTHORS	Ramkumar, Vidya; Shankar, Varsha; Kumar, Shuba

VERSION 1 – REVIEW

REVIEWER	Coco, Laura San Diego State University College of Health and Human Services
REVIEW RETURNED	12-Jun-2023

GENERAL COMMENTS	Thank you for the opportunity to read and review this paper. Overall this is a well-written paper and an interesting methodology to describe the factors influencing the sustainability of teleaudiology services. The qualitative information helps fill in the gaps that are not available from published studies. Below are my suggestions for the authors' consideration: Major: I strongly recommend updating this search to include more recently published articles. The number of telehealth publications has dramatically increased over the past few years, and these should be included if at all possible. In addition, recent research has moved towards adopting dissemination and implementation frameworks and therefore by updating the search, and it is possible that the authors will find papers published after April 2021 that include implementation outcomes. In the intro, please add more information and background on audiology, and teleaudiology. Introduce synchronous and asynchronous modes. Also add more background on implementation research and outcome frameworks. Introduce and describe IOF. Minor: State when the search was conducted. 1. In Table 1 - rather than Project # and the # of citation, write out the citation or abbreviated citation (First author's last name, year).2. (related to #1) I found the presentation of Project #s difficult to follow. Consider an alternative organization scheme (e.g., first author's last name, date).3, P 12 line 10: If keeping with the project number scheme, I recommend changing to P1-10; 14, 16, 18-20.4. P 12 line 15: Replace "few" with "six".5. P. 14 line 26: A more descriptive term for the professional-professional model is actually the professional-professional-patient
--

	model. A professional-professional model would involve something like interprofessional education via videoconference. 6. The organization of qualitative results should be improved. The authors refer the reader to quotes on another page which is cumbersome and disrupts the flow. Perhaps embed relevant quotes. 7. It is not clear why the sustainability section refers only to the scoping review and not the SSIs. 8. Please describe the project implementers (interviewees).
--	--

REVIEWER	Mutsvangwa, Tinashe University of Cape Town
REVIEW RETURNED	11-Aug-2023

GENERAL COMMENTS	I extend my gratitude to the authors for their significant contribution to the field of tele-audiology, which holds immense potential in bridging gaps in specialized services. The paper highlights advancements in tele-audiology over the past two decades, encompassing screening, diagnosis, rehabilitation, and the expansion facilitated by computerized equipment. The study appropriately identifies a stark disparity between promising research outcomes and the practical, sustainable implementation of tele-audiology in the real world. Despite its evident potential, there is a scarcity of data on successful, long-term implementations and the factors influencing sustainability. The authors admirably address this gap through a dual approach, employing a scoping review in line with PRISMA-ScR guidelines and qualitative semi-structured interviews. Their inclusive methodology, spanning a review of services from 2010 to 2021 across six databases and guided by the IOF framework, is commendable. Ethical considerations, including e-consent, are appropriately handled. The results are detailed, and the discussion is insightful, although it could provide more clarity on study limitations. The primary objective of identifying sustainable, long-term tele-audiology services beyond research and evaluating their sustainability, in alignment with StARI guidelines and factors outlined by the IOF, offers a valuable perspective. This comprehensive methodological framework underscores the urgent need for effective implementation strategies, continuous monitoring, and necessary adjustments to translate research into sustained tele-audiology practice. While the paper boasts overall strength, certain areas deserve further consideration: 1. The study's search strategy covered only six electronic databases, potentially excluding relevant studies from specialized fields or regions. The omission of grey literature, such as conference proceedings and theses, could result in an incomplete overview. Reliance on indexed databases may introduce publication bias, favouring positive outcomes and high-impact journals. Additionally, there is a possibility of language bias affecting the comprehensiveness of the scoping review. While no request for changes to the study or new results is made, addressing these potential limitations in the manuscript's discussion is advised.
---

	2. Justification for Timeline: The rationale behind selecting the timeframe of January 2010 to April 2021 requires clarification. 3. Elaboration on Thematic Analytical Approach: A more detailed explanation of the approach within the study's context is necessary. 4. Clarification of Sampling Approach: Providing a clear explanation of "purposive sampling" can help address concerns about representativeness. 5. Relevance of Authors' Gender: The rationale for reporting authors' gender, which may seem non-essential information in the study's context (Page 8, Line 49), should be explained. 6. Clarity in Data Collection: Defining "rich" data will enhance comprehension (Page 9, Line 1). 7. Explanation of the "A Priori Thematic Saturation" Model: Offering greater insights into the model's application will enhance transparency (Page 9, Line 10/11). 8. Addressing Sample Size and Thematic Saturation: The study's small sample size (six authors) should be acknowledged, and a more explicit description of the process to achieve thematic saturation is needed, along with an acknowledgment of potential selection biases.
--	--

VERSION 1 – AUTHOR RESPONSE

Reviewer: 1

Dr. Laura Coco, San Diego State University College of Health and Human Services

Comments to the Author:

Thank you for the opportunity to read and review this paper. Overall this is a well-written paper and an interesting methodology to describe the factors influencing the sustainability of teleaudiology services. The qualitative information helps fill in the gaps that are not available from published studies. Below are my suggestions for the authors' consideration:

Major:

I strongly recommend updating this search to include more recently published articles. The number of telehealth publications has dramatically increased over the past few years, and these should be included if at all possible. In addition, recent research has moved towards adopting dissemination and implementation frameworks and therefore by updating the search, and it is possible that the authors will find papers published after April 2021 that include implementation outcomes.

- As suggested by the reviewers and the editor, the search was updated to identify tele-audiology studies that fit the inclusion criteria until 2023 (page. 6, line 4). The updated search strategy is provided in appendix 2. While our literature search yielded a few interesting publications, none of them precisely fulfilled our inclusion criteria. However, one pertinent study was included in the discussion and two others were included in the introduction (page. 22, lines 6 – 8; page 4, lines 6 - 8). In the intro, please add more information and background on audiology, and teleaudiology. Introduce synchronous and asynchronous modes.

- The recommended change has been made in the introduction section of the manuscript (page 4, lines 2-3, 8-9).

Also add more background on implementation research and outcome frameworks. Introduce and describe IOF.

- The recommended change has been made in the introduction section of the manuscript (page 4, lines 11 – 19).

Minor:

State when the search was conducted.

1. In Table 1 - rather than Project # and the # of citation, write out the citation or abbreviated citation (First author's last name, year).

- Table 1's label has been changed to table 2 and it has been edited to include the citation (First author's last name, year) as recommended (page 11).

2. (related to #1) I found the presentation of Project #s difficult to follow. Consider an alternative organization scheme (e.g., first author's last name, date).

- We have chosen to maintain the existing project presentation system, since we believe that readability would be compromised if we were to adopt the format of "First author's last name, year," given the substantial volume of projects involved.

3. P 12 line 10: If keeping with the project number scheme, I recommend changing to P1-10; 14, 16, 18-20.

- The recommended changes have been incorporated throughout the manuscript (page 12, line 4 onwards).

4. P 12 line 15: Replace "few" with "six".

- The recommended change has been incorporated (page 12, line 5).

5. P. 14 line 26: A more descriptive term for the professional-professional model is actually the professional-professional-patient model. A professional-professional model would involve something like interprofessional education via videoconference.

- The recommended changes have been made throughout the manuscript (page 14, lines 4 and 12).

6. The organization of qualitative results should be improved. The authors refer the reader to quotes on another page which is cumbersome and disrupts the flow. Perhaps embed relevant quotes.

- We have chosen to maintain the existing flow of the qualitative results with the quotes in a separate table. Several qualitative studies use this format and we believe that readability would be compromised if we were to embed the quotes in the manuscript. It would also result in a significant increase in the word count, which strays from the manuscript guidelines.

7. It is not clear why the sustainability section refers only to the scoping review and not the SSIs.

- We reviewed the transcripts again and have added the relevant quotes from the semi-structured interviews to the sustainability section (page 19, lines 24 – 25; page 20, line 1).

8. Please describe the project implementers (interviewees).

- The recommended change has been incorporated and the project implementer characteristics have been summarized in table 1 for the reader's ease (page 8).

Reviewer: 2

Dr. Tinashe Mutsvangwa, University of Cape Town

Comments to the Author:

I extend my gratitude to the authors for their significant contribution to the field of tele-audiology, which holds immense potential in bridging gaps in specialized services. The paper highlights advancements in tele-audiology over the past two decades, encompassing screening, diagnosis, rehabilitation, and the expansion facilitated by computerized equipment.

The study appropriately identifies a stark disparity between promising research outcomes and the practical, sustainable implementation of tele-audiology in the real world. Despite its evident potential,

there is a scarcity of data on successful, long-term implementations and the factors influencing sustainability. The authors admirably address this gap through a dual approach, employing a scoping review in line with PRISMA-ScR guidelines and qualitative semi-structured interviews. Their inclusive methodology, spanning a review of services from 2010 to 2021 across six databases and guided by the IOF framework, is commendable. Ethical considerations, including e-consent, are appropriately handled. The results are detailed, and the discussion is insightful, although it could provide more clarity on study limitations.

The primary objective of identifying sustainable, long-term tele-audiology services beyond research and evaluating their sustainability, in alignment with StARI guidelines and factors outlined by the IOF, offers a valuable perspective. This comprehensive methodological framework underscores the urgent need for effective implementation strategies, continuous monitoring, and necessary adjustments to translate research into sustained tele-audiology practice.

While the paper boasts overall strength, certain areas deserve further consideration:

1. The study's search strategy covered only six electronic databases, potentially excluding relevant studies from specialized fields or regions. The omission of grey literature, such as conference proceedings and theses, could result in an incomplete overview. Reliance on indexed databases may introduce publication bias, favouring positive outcomes and high-impact journals. Additionally, there is a possibility of language bias affecting the comprehensiveness of the scoping review. While no request for changes to the study or new results is made, addressing these potential limitations in the manuscript's discussion is advised.

- We agree that there are certain limitations which could influence the scope of the current review such as the lack of a quality appraisal process and the possibility of non-English papers being missed. These limitations have been emphasized in the discussion section of the manuscript (page 24, lines 1 – 4).

- The lines "Conference proceedings and theses which did not report long-term tele-audiology implementation were also excluded" (page 6, lines 5 - 6) could have been misleading. However, we searched databases for grey literature (ProQuest and Google Scholar – already mentioned in page 5, lines 19 - 20) and only when conference proceedings and theses did not meet our inclusion criteria, we chose to exclude them. Hence, we have removed this statement from the manuscript.

2. Justification for Timeline: The rationale behind selecting the timeframe of January 2010 to April 2021 requires clarification.

- The successful implementation of health services necessitates a substantial duration of rigorous practice grounded in evidence-based approaches. During the early 2000s, tele-audiology was focussed on evidence building. Therefore, studies beyond 2010 were included during which period there was a greater possibility of implementation of tele-audiology services. As suggested by the reviewers and the editor, the search was updated to identify tele-audiology studies that fit the inclusion criteria until 2023 (page. 6, line 4). The updated search strategy is provided in appendix 2.

3. Elaboration on Thematic Analytical Approach: A more detailed explanation of the approach within the study's context is necessary.

- The purpose of the thematic analytical approach in the study's context has been clarified (page 7, lines 8 – 11).

4. Clarification of Sampling Approach: Providing a clear explanation of "purposive sampling" can help address concerns about representativeness.

- Being a qualitative study, representation is not possible; but we sought to obtain insights relevant to the study objectives. The word "purposive sampling" earlier used in the manuscript was incorrect as we approached all the project implementers identified through the scoping review. But only six

authors consented to participate. The participant selection has been reworded for clarity (page 7, lines 15 - 18).

5. Relevance of Authors' Gender: The rationale for reporting authors' gender, which may seem non-essential information in the study's context (Page 8, Line 49), should be explained.

- We added this information in adherence to the COREQ checklist. However, we understand that since the author's gender might be considered irrelevant, we have removed it from the main text and have summarized it in table 1 (page 8).

6. Clarity in Data Collection: Defining "rich" data will enhance comprehension (Page 9, Line 1).

- The term "rich data" is commonly used in qualitative research and refers to in-depth and complex information which allows the researcher to better interpret the meaning and context of findings (Saunders et al., 2018).

7. Explanation of the "A Priori Thematic Saturation" Model: Offering greater insights into the model's application will enhance transparency (Page 9, Line 10/11).

- The use of the model in the current study has been included (page 8, lines 12 – 13).

8. Addressing Sample Size and Thematic Saturation: The study's small sample size (six authors) should be acknowledged, and a more explicit description of the process to achieve thematic saturation is needed, along with an acknowledgment of potential selection biases.

- The recommended change has been incorporated (page 7, lines 21 – 23; page 8, lines 12 - 13; page 24, lines 1 - 4). Only six authors consented to participate in the SSIs and all of them were included. This has been emphasized in the discussion section of the manuscript (page 24, lines 1 - 4). There was no selection bias since all authors were contacted and those who consented were included.

VERSION 2 – REVIEW

REVIEWER	Mutsvangwa, Tinashe University of Cape Town
REVIEW RETURNED	27-Sep-2023
GENERAL COMMENTS	Thank you for addressing the comments I had.